# Gastrointestinal Microbiota of Spiny Lobster: A Review

**En Yao Lein [1], Mohammad Tamrin Mohamad Lal [1] , Balu Alagar Venmathi Maran [1] , Choon Looi Ch'ng [2], Katsuyuki Hamasaki [3], Motohiko Sano [3] and Audrey Daning Tuzan [1],***

[1] Borneo Marine Research Institute, Universiti Malaysia Sabah, Jalan UMS,
  Kota Kinabalu 88400, Sabah, Malaysia; my1921017t@student.ums.edu.my (E.Y.L.);
  mdtamrin@ums.edu.my (M.T.M.L.); bavmaran@ums.edu.my (B.A.V.M.)
[2] China Ocean Fishing Holdings Limited, Room 03, 22/F, China Resources Building, 26 Harbor Road,
  Wan Chai, Hong Kong; 13637547320@163.com
[3] Department of Marine Biosciences, Tokyo University of Marine Science and Technology, Konan 4-5-7,
  Minato-ku, Tokyo 108-8477, Japan; hamak@kaiyodai.ac.jp (K.H.); msano@kaiyodai.ac.jp (M.S.)
*  Correspondence: audrey@ums.edu.my; Tel.: +60-88-320000 (ext. 2587)

**Abstract:** The gastrointestinal (GI) microbiota is a group of complex and dynamic microorganisms present in the GI tract of an organism that live in symbiosis with the host and benefit the host with various biological functions. The communities of GI microbiota are formed by various aerobic, anaerobic, and facultatively anaerobic bacteria in aquatic species. In spiny lobsters, common GI microorganisms found in the GI tract are *Vibrio*, *Pseudomonas*, *Bacillus*, *Micrococcus*, and *Flavobacterium*, where the structure and abundance of these microbes are varied depending on the environment. GI microbiotas hold an important role and significantly affect the overall condition of spiny lobsters, such as secreting digestive enzymes (lipase, protease, and cellulase), helping in digesting food intake, providing nutrition and synthesising vitamins needed by the host system, and protecting the host against infection from pathogens and diseases by activating an immune mechanism in the GI tract. The microorganisms in the water column, sediment, and diet are primarily responsible for altering, manipulating, and shaping GI microbial structures and communities. This review also highlights the possibilities of isolating the indigenous GI microbiota as a potential probiotic strain and introducing it to spiny lobster juveniles and larvae for better health management.

**Keywords:** beneficial microbes; *Panulirus*; *Vibrio*; probiotics; aquatic invertebrate



## 1. Introduction

The gastrointestinal (GI) microbiota is composed of microorganisms mainly colonised by aerobic and anaerobic bacteria in the GI tract of an organism [1]. The GI microbiota engages in a symbiotic relationship with higher organisms in the GI tract and plays a supporting role and contributes to beneficial functions, including nutrition, digestion, the immune system, and the synthesis of vitamins [1–3]. Antonie Van Leeuwenhoek discovered bacteria when he found "strange little animals" microscopically, which led to the encounter of GI microbes in 1681 [4]. Nevertheless, the focus on GI microbiota research only began a few centuries later, when researchers discovered an anaerobic bacterium, *Escherichia*, in the human GI tract. With the advancement in biotechnologies and microbiology studies, scientists began to shift from human GI microbiota research to various kinds of terrestrial and aquatic life in order to understand the role of the GI microbiota in animals [4].

According to Wang et al. [5], the earliest studies on the intestinal and GI microbiota community in aquatic vertebrates date back to 1920. A review stated that the population of the GI microbiota in aquatic animals is lower compared to terrestrial animals and human beings [6,7]. This statement was later proved wrong by Zhou et al. [8], where the cultivable GI microbes were less than 0.1% of the total microbial community in the GI tract of fish. The trend of studying fish GI microbiota is still ongoing, and the current focus is on the

application of GI microbes in aquaculture, such as prebiotics and probiotics [5]. These applications aim to improve fish health, boost the fish immune system, and enhance food security [5,9–11].

The presence of microorganisms in the digestive tract of invertebrates has long been recognised, where gut microbes are found in almost every invertebrate, including Annelida, Echinoidea, Mollusca, aquatic detritivores, corals, and crustaceans [12,13]. Like any other terrestrial animal and aquatic vertebrates, crustaceans contain a wide diversity of indigenous GI microbiota, which is vital in maintaining the host immune system, digestion, enzymatic reactions, nutrition, and better growth performances [14,15]. Based on a review [12], an earlier example survey on the intestinal microbiota community was revealed [16,17]. Conclusively, the predominant genera found in the intestinal tract of crustaceans were *Vibrio* spp., *Pseudomonas* spp., and *Coryneforms* spp. [16]. This statement is supported by recent findings on the microbial community in the intestinal tract of different crustaceans, such as prawns, shrimps, crabs, and lobsters [15,18–21].

## 2. Overview of GI Microbiota in Spiny Lobsters

The presence of microflora in the intestine of lobsters was reported in the early 1970s [22]. Upon finding various microflora in the stomach and intestine of the European lobster *Homarus vulgaris* (H. Milne-Edwards, 1837), the researchers proceeded to investigate the bacterial flora in the internal organs responsible for an infectious disease known as Gaffkaemia. The intestine of the European lobster consists of the bacterial community, fungi community, and some yeast-like organisms [22]. Further studies have isolated and identified a wide variety of intestinal microorganisms in European lobsters [23], which included *Micrococcus* spp., *Sarcina* spp., *Candida* spp., *Brevibacterium* spp., *Bacillus* spp., *Paracolon* spp., *Flavobacterium* spp., *Achromobacter* spp., *Pseudomonas* spp., and *Vibrio* spp. [23].

Extensive studies of intestinal microflora in the late 1980s were conducted in which eight strains of Gram-positive cocci were isolated from the gut of the spiny lobster *Panulirus japonicus* (von Siebold, 1824). The strains were unable to grow in complete aerobic conditions, but they possessed similar morphology and characteristics: they were Gram-positive, non-spore-forming cocci, and they grew under anaerobic conditions. In the latter, decreasing oxygen tension and unchanged carbon dioxide tension conditions needed to be fulfilled for the strains to continue growing. The biochemical test results and a comparison of the overall characteristics enabled the strains to be identified as Gram-positive cocci and anaerobic *Streptococcus* spp. [24]. Further studies analysed the gut microflora of marine crustaceans, the Japanese spiny lobster *P. japonicus*, and six coastal crabs: *Atergatis floridus* (Linnaeus, 1767), *Schizophrys aspera* (H. Milne-Edwards, 1831), *Tiarinia cornigera* (Latreille, 1825), *Pachygrapsus crassipes* (Randall, 1839), *Thalamita prymna* (Herbst, 1803), and *Plagusia dentipes* (De Haan, 1835) [16]. A total of 720 strains were isolated from the gut of the Japanese spiny lobster with *Vibrio* spp. as the dominant group (600 strains), followed by *Pseudomonas* (20 strains), *Staphylococcus* (40 strains), *Coryneforms* (20 strains), and anaerobic Gram-positive cocci (40 strains). These isolated strains differed from the microbial composition [16] reported in a previous study [23], and factors such as different host species, diets, surrounding environment, and temperature may explain the discrepancies. Sugita et al. [16] also found that the sampling time and temperatures had significant effects on crustacean gut microbiota populations. The results suggest that the gut microbial community in crustaceans may not be as stable as in marine fishes, where the surrounding water temperature easily influences the gut microbial community.

Further exploration of gut microflora in the intestinal tract performed by Ueda et al. [25] led to the isolation of more types of bacteria. As expected, *Vibrio* was the dominant genera, followed by *Pseudomonas*, *Flavobacterium*, *Micrococcus*, *Staphylococcus*, and *Corynebacterium*. More importantly, *Flavobacterium* and *Micrococcus* were newly discovered in the gut of spiny lobsters. The bacteria *Streptococcus* was also successfully isolated from the gut of the spiny lobster under obligate anaerobic and facultative anaerobic conditions [25]. While comparing the GI microflora of the spiny lobster with other coastal animals (crustaceans,

gastropods, and fishes), *Vibrio* was found to be dominant in the GI tract of all coastal animals. Nevertheless, different *Vibrio* spp. were dominant in different types of coastal animals. Thus, Ueda et al. [25] concluded that the microflora diversity in the intestinal tract is specific to each type of animal [25].

As the spiny lobster has become an important component in the fish trade, the import and export trade has drastically increased in the 20th century. Thus, the intestinal bacterial diversity in spiny lobsters during the transportation process has been investigated. Immanuel et al. [14] isolated and identified the intestinal bacteria of an unpacked spiny lobster *P. homarus* (control) and a packed spiny lobster with a time interval of 2 to 14 h. The intestinal bacterial loads and compositions of the unpacked spiny lobster (control) were significantly different in the samples isolated by the researchers. In contrast, the intestinal bacterial load and composition of the comparable samples depicted no significant difference. The bacterial species found in the spiny lobster intestine were *Pseudomonas aeruginosa* (dominant), followed by *Vibrio parahaemolyticus*, *Bacillus circulans*, *Escherichia coli*, *Photobacterium damselae*, *Flavobacterium columnar*, and *Micrococcus luteus*. Given that the spiny lobster has become a prospective aquaculture species, most spiny lobster intestinal research has shifted to aquaculture-related fields to overcome the challenges in the culture of spiny lobsters. These studies aimed to understand the microbial community in the aquaculture system of spiny lobster larvae; the application of probiotics, prebiotics, and synbiotics for lobster culture improvement; and the isolate-potential probiotics for spiny lobsters [26–29].

## 3. Development of GI Microbiota in Spiny Lobsters

The microbiota forms a diverse and complex ecological environment in the GI tract of lobsters. It is suggested that the development of spiny lobster GI microbiota could be influenced by several factors, such as lobster eggs, larval rearing system, aquatic environment, and feed intake. In an earlier exploration of the bacteria of lobster eggs, a filamentous bacterium, *Leucothrix mucor*, was found attached to the eggs of aquatic crustaceans, causing severe infections and high mortalities [30–32]. In the early 20th century, Gil-Turnes and Fenical [33] discovered a single, rod-shaped, Gram-negative epibiotic bacterium covering the lobster egg with a mosaic-like pattern, which is believed to protect against infection by the fungi *Lagenidium* spp. Bacteria from the family *Vibrionaceae* are some of the most common bacteria in the GI tract of lobster. The discoveries of *Photobacterium* in the eggs of spiny lobsters suggest that the GI microbes may develop within the eggs carried on the pleopods of the female spiny lobster [34]. The microbial isolates from the early phyllosoma stages of reared spiny lobsters consist of *Vibrio* spp., *Photobacterium* spp., and *Pseudomonas* spp. [35]. According to Bourne et al. [18], histology results revealed no sign of bacteria proliferating on the first day after hatching, as most bacteria colonising the gut of phyllosomas were only seen on the 20th day post-hatching. The microbial community present in the water column of the spiny lobster aquaculture system could directly affect the microbial community inside phyllosomas. The bacteria isolated from the water column of the spiny lobster phyllosoma aquaculture tank mainly comprised *Vibrio* spp., followed by *Pseudomonas* spp., *Photobacterium* spp., and *Bacillus* spp. [18,26]. A filamentous bacterium was observed in the live and late dead stages of phyllosomas via Scanning Electron Microscope (SEM). By using fluorescent in situ hybridisation (FISH), the filamentous bacteria were identified as *Thiothrix* spp. The presence of *Thiothrix* spp. may potentially affect the phyllosoma's moulting process and weaken its immune ability in the early stages [35].

The microbial community of wild phyllosomas shows significant differences in bacterial loads and diversity compared to reared phyllosomas [36]. There was no sign of filamentous bacteria or infectious symptoms in wild phyllosomas, and the isolated bacteria were also different compared to reared phyllosomas. This contrasting result indicates that microbial diversity is influenced by the surrounding environment [15,36,37]. The gut microbiota diversity is more complex and dynamic in juvenile spiny lobsters. For juvenile spiny lobsters, the intestinal microbiome found were phyla Mollicutes, Gammaproteobac-

teria, Aphaproteobacteria, Saprospirae, Bacteroidia, Deltaproteobacteria, Actinobacteria, and Flavobacteria [38]. A study conducted by Meziti and Kormas [39] on the gut bacteria community of the Norway lobster reflected a similar result, which supports the findings of Ooi et al. [38].

## 4. Composition of GI Microbiota in Spiny Lobsters

The microbiome composition of aquatic species is diverse and specific according to their surrounding environment and feeding habits. The microbiota found in the GI tract of aquatic species includes bacteria, fungi, viruses, and yeast [1,40]. According to Talwar et al. [41], approximately $10^8$ bacterial cells, which cover over 500 species of bacteria, inhabit the GI tract of fish. Aerobic, facultatively anaerobic, and obligate anaerobic bacteria are the common bacteria colonising the GI tract of fish [7,41]. With the help of next-generation sequencing (NGS), research has shown that the composition of fish GI microbiota comprises Proteobacteria, Fusobacteria, Firmicutes, Bacteroidetes, Actinobacteria, and Verrucomicrobia [5]. Meanwhile, the major bacteria phylum colonising the GI tract in aquatic crustaceans is Proteobacteria (Gram-negative, facultative, or obligate anaerobic bacteria) [42]. Specifically, the bacteria load present in the GI tract of the spiny lobster ranges from $10^6$ to $10^9$ per gramme of GI tract. The bacteria under the phyla Proteobacteria and Bacteroidetes were commonly found in the GI tract and haemolymph of various lobster species, including *P. ornatus* (Fabricus, 1798), *P. versicolor* (Latreille, 1804), *P. japonicus*, *P. homarus* (Linnaeus, 1758), *Homarus americanus* H. Milne-Edwards, 1837, and *H. gammarus* (Linnaeus, 1758) [14,15,25,43–45]. Previous studies also shows that bacteria communities isolated from GI tract of lobster are varies according to region (Table 1).

Apart from bacteria, fungi were found to inhabit the GI tract of aquatic species. Nevertheless, there are limited studies on the GI interaction of fungi and bacteria communities in aquatic species. Fungi are the key decomposers in the ecosystem, and they regenerate in a wide variety of places, such as substrata, wood, sand, algae, coral, and other living things [46,47]. Fungi usually engage in commensal, mutualistic, and parasitic relationships with other aquatic species [48]. The association between the parasite and pathogen forms is most prevalent in marine organisms [48]. Marine fungi that cause diseases to fish and shellfish can be grouped into oomycetes (water mould): *Lagenidium* spp., *Haliphthoros* spp., *Halocrusticida* spp., *Halioticida* spp., *Atkinsiella* spp., and *Pythium* spp.; and diasporic fungi: *Fusarium* spp., *Ochroconis* spp., *Exophiala* spp., *Scytalidium* spp., *Plectosporium* spp., and *Acremonium* spp. [49]. The common fungal pathogens affecting lobsters are *Lagenidium* spp., *Haliphthoros* spp., and *Halocrusticida* spp., which infect eggs and larvae, and *Zasmidium musae*, which causes significant problems in lobster seed production [48,50].

Recent studies discovered that *Penicillium* spp. and *Aspergillus* spp. produce antibacterial and antifungal compounds, which are suggested to be responsible for their pathogenicity [51]. A previous study revealed that *Penicillium* spp. were isolated from the GI tract of wild and reared spiny lobsters [45]. Besides the antimicrobial properties of *Penicillium* spp., the researchers found that the *Penicillium* strain can produce a low concentration of $\alpha$-amylase, which helps in indigestion [45]. Similarly, a recent study discovered that the fungal genus *Malassezia* was ecologically hyper-diverse [52]. *Malassezia* is a dominant component in human skin niches (both healthy and disease), and it has been found in various habitats, such as deep-sea sediments, corals, the GI tract of Japanese eels, the GI tract of lobsters, the exoskeleton of shellfish, and nematodes [53]. Yeast is also often found in healthy vertebrate and invertebrate aquatic species, comprising those from indigenous and terrestrial environments [54]. Meanwhile, *Rhodotorula*, *Candida*, *Debaryomyces*, and *Cryptococcus* were the least significant yeast genera isolated from aquatic animals [55]. The proliferation of yeasts in fish mucus is normally considered commensalism, where only a few pathogenetic cases are reported [56].

Yeast can also be found abundantly in the GI tract and the internal fluids of marine invertebrates, such as crab, shrimp, lobster, and sponges [48]. In the case of spiny lobsters, the preliminary study by Volz et al. [57] isolated yeast from the intestine while assuming

the presence of yeast was due to the accidental ingestion of yeast from the environment, which did not cause any pressure to the hosts. The findings were supported by Kumar et al. [45], who isolated a yeast strain with probiotic properties in the midgut of wild and reared spiny lobsters in a range of $10^2$–$10^3$ colony-forming units (CFU). Yeast was also found to have the ability to colonise the GI tract of aquatic animals, supply vitamins to the host, and help in immune response [55,56].

Marine viruses are transmitted into the host via infection in more than 90% of microorganisms present in the marine system and often cause mortality and diseases to the host [58]. Viral communities can regulate host population dynamics and participate in the biogeochemical cycle and carbon sequestration in the marine environment [59]. Viruses inhabiting the GI tract of spiny lobsters are often pathogenic. For instance, PaV1 (*Panulirus argus* virus 1) and WSSV (white spot syndrome virus) are viruses in the GI tract of spiny lobsters. PaV1 was first discovered in the juvenile Caribbean spiny lobster *P. argus* (Latreille, 1804) in Florida in 1999 [60]. PaV1 is also currently the only naturally occurring pathogenetic virus affecting various lobster species [61]. The pathogen, infected haemocytes, and spongy connective tissue cells, including hepatopancreases, GI tract, heart, gills, and ovary tissues, were identified using fluorescence in situ hybridization (FISH) [62]. Zamora-Briseno et al. [63] reported that spiny lobsters infected with PaV1 have a higher *Vibrio* composition compared to healthy spiny lobsters, which suggests that PaV1 has the ability to alter GI tract microbial composition. Alongside PaV1, a non-naturally occurring virus known as WSSV was also discovered in the GI tract and infected *Panulirus* spp. and *Homarus* spp. [64–66]. WSSV is a double-stranded DNA from the family Nimaviridae. The virus infects almost all crustacean species and causes mass mortalities of aquaculture species, leading to significant impacts on global crustacean aquaculture industries [61,67]. Given that WSSV is non-naturally occurring and highly infectious to most crustacean species, the potential emergence of WSSV into the wild environment may disrupt ecosystem balance and cause fisheries loss [61].

**Table 1.** Bacteria community in the GI tract of lobsters from year 1972–2020.

| Lobster | Country | Bacteria | References |
|---|---|---|---|
| *Homarus americanus* | Canada | *Micrococcus, Sarcina, Candida, Brevibacterium, Bacillus, Paracolon, Flavobacterium, Achromobacter, Pseudomonas*, and *Vibrio* | [23] |
| *Panulirus japonicus* | Japan | Streptococcus | [24] |
| *Panulirus japonicus* | Japan | *Vibrio* spp., *Pseudomonas* spp., *Staphylococcus* spp., and *Coryneforms* spp. | [16] |
| *Panulirus japonicus* | Japan | *Vibrio, Pseudomonas, Flavobacterium, Micrococcus, Staphylococcus, Corynebacterium, Streptococcus*, and *Bacteroidaceae* | [25] |
| *Panulirus homarus* | India | *Pseudomonas Aeruginosa, Vibrio parahaemolyticus, Bacillus circulans, Escherichia coli, Photobacterium damselae, Flavobacterium columnare*, and *Micrococcus luteus* | [14] |
| *Panulirus ornatus* | Australia | *Vibrio, Photobacterium*, and *Pseudomonas* | [35] |
| *Panulirus ornatus* | Australia | Mollicutes, Gammaproteobacteria, Alphaproteobacteria, Saprospirae, Bacteroidia, Deltaproteobacteria, Antinobacteria, and Flavobacteria | [38] |
| *Panulirus homarus* | India | *Enterobacter, Acinetobacter, Bacillus, Vibrio, Pseudomonas, Micrococcus*, and *Moraxella* | [15] |
| *Homarus gammarus* | Cornwall, UK | *Vibrio, Synechococcus, Spongiimonas, Spirochaeta, Shewanella, Roseovarius, Psychromonas, Psychrilyobacter, Photobacterium, Kiloniella, Candidatu, Arcobacter, Allofrancisella*, and *Aliivibrio* | [44] |
| *Panulirus argus* | Mexico | *Vibrio, Sphingomonas, Cetobacterium, Candidatus Hepatoplasma*, and *Candidatus* | [63] |

## 5. Role of GI Microbiota in Spiny Lobsters

Indigenous intestinal microbiota influences the physiological function of the host GI tract. These influences are evident in scientific articles reporting intestinal microflora

abnormalities in a variety of organisms, including germ-free animals, animals, and humans [68,69]. As observed in higher organisms, humans, terrestrial animals, or aquatic animals, the presence of GI microbiota in spiny lobsters influences the host and brings various benefits to its host [1]. The symbiosis between the GI microbiota and the host is important to ensure the nutrition and health of the host [70]. The human GI tract comprises 1000 culturable microflora species, where 92 species are from eukarya, 8 from archaea, and 957 from bacteria. The complex and diverse microflora community in the GI tract of humans co-exists and contributes to the prevention of metabolic diseases [4]. The GI microbiota in the human body possesses both beneficial and harmful traits [71]. It has been proven that GI microbes are associated with obesity, inflammatory bowel disease, cancers, diabetes, autism, and asthma [72–76]. The role of fish GI microbiota is based on the host's dietary needs. Herbivorous fishes, such as grass carp, are associated with cellulolytic bacteria, which help in plant fibre intake. Meanwhile, nitrogen-fixing bacteria are the dominant species in the GI tract of wood-eating fish [77,78]. The GI bacteria of carnivorous fish species consist of mostly lipase- and protease-synthesising bacteria [77,78]. In aquatic invertebrates, an earlier experiment by Harris [13] proved that the presence of GI bacteria has specific roles and functions.

### 5.1. Role of GI Microbiota in Digestion

Spiny lobsters feed on live fish, molluscs, other crustaceans, aquatic worms, and some aquatic plants; hence, they are categorised as omnivores. Shrimps and prawns are also classified as omnivores, as their GI microbiota is similar to that of spiny lobsters. The GI tract of spiny lobsters is colonised by bacteria such as *Vibrio*, *Pseudomonas*, *Aeromonas*, *Pseudoalteromonas*, *Photobacterium*, and *Plesiomonas* [13,17,79,80]. Studies on the digestive relevance of GI microbiota are mostly conducted on freshwater and seawater fish, while crustaceans, especially spiny lobsters, have received little attention. The microbiota present in the GI tract of spiny lobsters mostly consists of proteolytic bacteria, amylolytic bacteria, lipolytic bacteria, and cellulolytic bacteria [45,81]. Other bacteria, such as the *Flavobacterium* found in the GI tract of spiny lobsters, are capable of hydrolysing complex polysaccharides [82,83]. All these bacteria are essential to spiny lobsters, as they help in producing enzymes to digest lipase, protease, amylase, and chitin. Spiny lobsters attach to their carnivorous feeding preferences (molluscs, crustaceans, polychaete worms, and echinoderms), which form high proteolytic enzyme activities (trypsin, chymotrypsin, and carboxypeptidase A) and low lipase activities in the digestive tract [84]. Kumar [45] compared the microbial diversity in the intestine of wild and laboratory-reared spiny lobsters *P. versicolor*. The researchers found that cellulolytic bacteria were dominant in the foregut of normal spiny lobsters, while proteolytic bacteria were dominant in the foregut of laboratory-reared spiny lobsters. These results indicate that the feeding behaviour of spiny lobsters could change the enzyme-synthesising bacteria in the GI tract.

### 5.2. Role of GI Microbiota in Nutrition

Indigenous GI microbiotas possess important roles in the well-being of their host and contribute to nutrient acquisition in humans, terrestrial animals, and aquatic animals [85,86]. The ability of GI-associated microbes to digest and synthesise vitamins from daily diets has mostly been documented in human and animal studies, whereas limited information is available regarding insects, fish, and aquatic invertebrates [68,85,87,88]. More than 90% of the mammalian population on the planet are herbivores, and these animal species are unable to produce enzymes to digest carbohydrates and cellulose. Thus, GI microbes are crucial in the degradation and digestion of food [89].

The GI bacteria ferment dietary carbohydrates into short-chain fatty acids (SCFA), which support the host by providing energy and facilitating the absorption of Sodium (Na) and water into cell tissues. Cellulolytic bacteria present in the GI tract assist the host to completely hydrolyse cellulose into glucose, which is the compound available to the host. This process involves the action of exoglycanases, endoglucanases, and β-glucosidases [90].

The host absorbs protein in amino acids degraded by the proteolytic bacteria in the GI tract. The synthesis of B vitamins (complex of 10 water-soluble compounds) is also facilitated by GI microbes. This process is well-documented in fish, where the amount of vitamin B12 varies according to species [91,92]. Based on Nayak [86], the production of vitamin B12 is closely related to the abundance of anaerobic bacteria compared with aerobic bacteria in the GI tract of fish. The GI microbiota is also involved in nutrient-material uptake stimulation, especially in cholesterol metabolism and trafficking in aquatic animals [5,93]. The use of gnotobiotic models demonstrated that germ-free zebrafish larvae failed to degrade and absorb proteins in the intestine, but these functions were performed effectively in the later development stages following the enrichment of the GI microbiota [94]. To date, the nutritional role of the GI microbiota in spiny lobsters remains unclear. The GI microbiota–host interaction could be better understood by utilising gnotobiotic models. Previous studies on gnotobiotic daphnia and gnotobiotic artemia provide a possibility for a better understanding of the interaction and the importance of GI microbiota in spiny lobsters [95].

### 5.3. Role of GI Microbiota in Immune System

Gut-associated lymphoid tissues (GALTs) are GI microbiota responsible for the immune system, in which intestinal microbiota is important for the complete development of mature immune cells [96,97]. The intestinal epithelium will secrete and be covered by a layer of mucus, which acts as the first line of defence against harmful microbes. The symbiosis between GALT, mucus, and indigenous intestine microbes will mature the host's gut-associated immune system [86]. According to Lee and Mazmanian [98], the approach on germ-free animals provides a better understanding of the interaction between host GI microbes and the immune system, where the protection from intestinal mucosa is defective in germ-free mice. The gut-associated immune system mechanism (Payer's patches, lymph nodes, and lamina propria) is smaller and inactive in germ-free animals, while exposure to antigenic stimuli could reverse the action [98–100].

The functions of T cells in protecting the host from various infections are also promoted by the GI microbiota. For instance, *Bacteroides fragilis* promotes T cells to protect against *Helicobacter hepaticus* infection, while *Bifidobacterium infantis* diminishes *Salmonella typhimurium* intestinal infection [99]. More than 70% of the total body of Immunoglobin A (IgA) across the mucosa membrane surface is secreted by the GI microbiota, especially in the intestine [101–103]. IgA is an important component of first-line defence and interacts with specific receptors and immune mediators for protective functions [104].

Intestinal immunity in aquatic animals is less advanced compared to terrestrial animals. Nonetheless, aquatic animals are exposed to higher microbial infection challenge, as they inhabit a microbial-rich environment [105,106]. The immunity-associated mechanism of fish is composed of gut-associated lymphoid tissue (GALT), skin-associated lymphoid tissue (SALT), gill-associated tissue (GIALT), and nasopharynx-associated lymphoid tissue (NALT), which were recently uncovered [106]. According to Gomez and Balcazar [2], the gut-associated immune system of aquatic vertebrates and terrestrial vertebrates are similar in that the epithelial cells and mucosa act as a selective barrier and promote T cells and B cells to produce IgA as an intestinal defence mechanism. Indigenous GI microbiota may suppress the foreign microbiota and prevent the colonisation and proliferation of pathogens through the colonisation-resistance process [107]. The indigenous microbes secret and release antimicrobial peptides to win over the competition between the pathogen and the niche space [108].

The immune mechanism in aquatic invertebrates is different compared to vertebrates. In crustaceans, the hard exoskeleton, made of cuticle, acts as the first line of defence against microbial infection [109]. Conversely, when pathogens successfully invade the body or the tissue of invertebrates, the innate immune system will instantly activate and eliminate the intruding pathogen [110]. The innate immune system differs between invertebrates and vertebrates, as evidenced in the lack of antibodies and the critical molecular and cellular players, such as B lymphocytes and T lymphocytes, in aquatic invertebrates [111].

The innate immune system of invertebrates is classified into two groups: humoral immunity (haemolymph agglutination, prophenoloxidase system (proPO), and antimicrobial peptides) and cellular immunity (phagocytosis, encapsulation, and haemocyte nodulation) [112–114]. There is a paucity of data regarding the role of GI microbes in the immune system of invertebrates. A recent study found that the gut microbiota of shrimp plays a vital role in maintaining host health, and the colonisation-resistance process may occur in the digestive tract of shrimp [42]. The research conducted by Tepaamorndech et al. [115] revealed that the introduction of the bacteria strain *Bacillus aryabhattai* into the GI tract of shrimp could suppress the population of *Vibrio* spp. and stimulate innate immunity and antioxidant activities.

## 6. Factors Influencing GI Microbiota in Spiny Lobsters

The complex and diverse microbiota communities that inhabit the GI tract are different and unbalanced in every species due to other external factors. Factors such as environmental, geographical, dietary, genetic, and disease infection are responsible for shaping the GI microbiota communities of an organism [44,63,116–118]. Sullam et al. [3] demonstrated the influence of water salinity, trophic level, and host phylogeny on GI microbial composition. The researchers found similarities in the gut microbiota composition between herbivores, fishes, and mammals. Wild-captured and domestic-cultured black tiger shrimps *Penaeus monodon* (Fabricius, 1798) shared some bacterial members. Conversely, differences in habitat provide evidence of internal environmental pressure in selective intestinal bacteria [119].

### 6.1. Environmental Factors

Microbial loads in the aquatic environment are richer than in terrestrial, air, and soil, where aquatic microbes significantly affect the formation of GI microbes in an aquatic organism [120]. Aquatic environmental factors, such as temperature, pH, salinity, chemical oxygen demand (COD), and dissolved oxygen (DO), significantly impact the abundance of the GI microbiota community [121]. Sun et al. [121] isolated and compared the bacteria from the water column, sediment, and intestine of different aquatic species. Resultantly, the significant phyla were *Proteobacteria*, *Bacteroidetes*, *Cyanobacteria*, *Acidobacteria*, *Actinobacteria*, and *Firmicutes*. The findings indicated that the primary sources of intestinal bacteria include the surrounding water column, sediment, and the type of environments, such as freshwater and seawater. These sources demonstrated significant impacts on the microbial composition in the intestine. Another study on the intestine of oriental river prawns documented three significant phyla, with Proteobacteria being the dominant (23–60% of the total population), followed by *Firmicutes* and *Actinobacteria* [122]. Concurrently, in a seawater environment, Proteobacteria typically make up more than 80% of the total population of microbes in the intestine of oriental river prawns in a seawater environment, followed by *Firmicutes* and Bacteroidetes [122].

Previous studies have also focused on the intestinal bacteria communities of wild-caught and domesticated-breed *P. monodon*. The major phyla inhabiting the GI tract involved *Proteobacteria*, *Bacteroidetes*, *Firmicutes*, and *Actinobacteria*, with *Proteobacteria* being the most abundant phyla in both wild-caught and domesticated-breed *P. monodon* [18,123]. Wild-caught and domesticated-breed *P. monodon* shared similar types of bacteria but differed in terms of bacterial richness [122]. Accordingly, wild-caught *P. monodon* was significantly richer than domesticated-breed [122]. The microbial phyla found in the intestine of crayfish from Jingzhou, Yangzhou, and Xuyi included *Tenericutes*, *Bacteroidetes*, *Firmicutes*, *Proteobacteria*, and RsaHF231. Crayfish samples collected from Yangzhou demonstrated a higher abundance of *Firmicutes* than those collected from Jingzhou and Xuyi, thus reflecting that the development of intestinal microbiota may be influenced by geographical location [118].

Holt et al. [44] demonstrated the impact of spatial and temporal axes on the gut microbiome in European lobsters. The GI bacteria community in sea-based container culture (SBCC) lobsters demonstrated significantly higher diversity compared to land-

based culture (LBC) lobsters. Specifically, the major genera found in SBCC were *Vibrio*, *Spongiimonas*, *Candidatus Hepatoplasma*, *Aliivibrio*, and *Photobacterium*, while in LBC, they were *Carboxylicivirga*, *Arcobacter*, *Psychrilyobacter*, *Candidatus Hepatoplasma*, and *Vibrio*. The structure and composition of GI microflora of wild spiny lobsters were also different based on their geographical locations. Wild spiny lobsters caught from the coastal area of India had intestinal microflora made up of genera *Enterobacter*, *Acinetobacter*, *Bacillus*, *Vibrio*, *Pseudomonas*, *Micrococcus*, and *Moraxella*. In contrast, spiny lobsters caught from water around Mexico formed GI microflora with genera *Vibrio*, *Sphingomonas*, *Cetobacterium*, *Candidatus Hepatoplasma*, and *Candidatus* [15,63]. The differences in GI microflora composition may be due to genetic background, biological habits, feeding background, water properties, and seasonal impact [5,121,124].

*6.2. Dietary Factor*

Dietary habits and nutrition intake significantly influence the composition and structure of the GI microbiota [122]. Protein intake, lipids, probiotics, and prebiotics from diets are essential in the manipulation of GI microbes [1]. Several studies have uncovered the relationship of diets with intestinal microbiota in swimming crabs, abalones, shrimps, crayfish, and spiny lobsters [63,80,120,121,125]. The abundance of microbial communities in the GI tract could be altered by changing the dietary lipid level (Table 2). Proteobacteria were dominant in the GI tract of juvenile swimming crabs *Portunus trituberculatus* (Miers, 1876) fed with medium- and high-fat diets, whereas phyla Fusobacteria were dominantly fed low-fat diets [21]. The type of lipid source was also recorded as altering the composition of bacteria in the intestine of Pacific white shrimps [80]. Pacific white shrimps fed with soya oil (SO) and beef tallow (BT) reflected a lower Rhizobiaceae count in the intestine than shrimps fed with a combination diet of linseed oil, SO, and BT (SBL). Pacific white shrimps fed with SO and BT recorded more Aeromonadaceae and Enterobacteriaceae than shrimps fed with SBL.

**Table 2.** Dietary lipid level shaping GI tract microbial communities [21].

| Diets | Phyla | Families | Relative Abundance % | Dominant |
|---|---|---|---|---|
| High-fat diet (15.1%) | Proteobacteria | Vibrionaceae | 41.8 | Vibrionaceae |
| | Fusobacteria | Leptotrichiaceae | 1.1 | |
| | Tenericutes | Mycoplamataceae | 19.6 | |
| Medium-fat diet (9.9%) | Proteobacteria | Vibrionaceae | 67.4 | Vibrionaceae |
| | Fusobacteria | Leptotrichiaceae | 9.5 | |
| | Tenericutes | Mycoplamataceae | 20.0 | |
| Low-fat diet (5.8) | Proteobacteria | Vibrionaceae | 7.0 | Liptotrichiaceae |
| | Fusobacteria | Leptotrichiaceae | 60.8 | |
| | Tenericutes | Mycoplamataceae | 29.3 | |

Meziti et al. [39] reported that the relationship of the Norway lobster's gut microbes with diets and tank water was insignificant. Bacterial diversity in starved Norway lobster samples was higher than in samples fed with frozen mussels and formulated pellets. Bacteria in the GI tract of squat lobsters vary according to their feeding habits [126]. For instance, squat lobsters fed on wood were colonised by cocci, small-rod, spiral-shaped, chain-shaped, and filamentous bacteria, while those fed on whale-bone-containing boxes, wood, and turtle-shell substrates were colonised by both small and long fusiform-rod, thin-rod, and relatively low-cocci bacterial [126].

Mannan oligosaccharide (MOS) is one of the popular dietary supplements that boost the growth and health of marine species, such as fish, shrimps, and lobsters. *Vibrio* spp. are some of the most common bacteria found in lobsters, crabs, and shrimps due to their essential role in nutrient cycling; however, some *Vibrio* spp. are pathogenic and promote the transmission of diseases [127,128]. Juvenile spiny lobsters (*P. ornatus*) fed with the MOS diet have 10 times higher aerobic bacteria in the gut than those lobsters fed with trash fish [129].

Likewise, the population of *Vibrio* spp. was approximately seven times higher than in lobsters fed with trash fish. Spiny lobster (*P. homarus*) juveniles fed with MOS significantly reduced the number of *Vibrio* spp. in the GI tract compared to formulated diets and bycatch diets, which is different from the findings of Sang and Fotedar [129,130]. European lobsters (*H. gammarus*) fed with MOS and *Bacillus* spp. demonstrated a decreasing population of *Vibrio* spp. in the GI tract. Although a *Bacillus* spp. + MOS diet significantly improved the overall condition of the lobsters, the researcher found that the richness and diversity of GI microbiota were reduced [27]. Future studies should consider developing a method to improve the overall condition of spiny lobsters without affecting the species richness and diversity of their GI microbiota.

## 7. Application of Gastrointestinal Microbiota

Microbiotas inhabiting the GI tract of living organisms possess numerous benefits, such as enhancing the immune system, secreting digestive enzymes, and promoting nutrition. The significant application of GI microbiota is in the synthesis of probiotics [131]. The concept of probiotics was first introduced by Liley and Stillwell in 1965, where the word "probiotic" was derived from the combination of two Greek words, "pro" and "bios", which means "for life" [132]. Apart from being applied in probiotics, GI microbiotas are employed in various sectors in the livestock, food, beverage, and pharmaceutics industries [133]. GI microbiotas, such as lactic acid bacteria (LAB), are often used as food preservatives in the food and beverage industries due to their ability to synthesise bacteriocin, which inhibits the growth of foodborne pathogens [134]. LAB is also an essential component in shaping the flavour and texture of food through the fermentation process and preventing food spoilage [134]. Other than LAB, yeast (one of the symbionts in the GI tract) is also important in making alcoholic products, baking, and dairy product processes [134]. In industrial sectors, the enzymes synthesised by the GI microbiota (cellulase, lipase, and amylase) are used in pulp and paper making, textile making, detergent, and cosmetics [134]. Bifidobacterium and *Lactobacillus* are the major groups of bacteria colonising the GI tract and are often isolated as probiotics [135]. According to Sreeja and Prajapati [136], Bifidobacterium and *Lactobacillus* are often used in pharmaceutical sectors and are clinically proven to affect the prevention of diarrhoea, relieve constipation, improve the immune system, and reduce abdominal bloating.

The application of GI microbiota as a probiotic is also widely used in finfish and shellfish aquacultures as an alternative to antibiotics [131,132,137,138]. Spiny lobsters are one of the most promising aquaculture species due to their high market price and stable market demands. The use of probiotics in the spiny lobster aquaculture aims at a closed-life cycle production and developing an economically viable method of raising lobsters from eggs through market size. The earlier record of using probiotics in lobsters was conducted by Daniels et al. [27], in which commercial probiotics (*Bacillus* spp.) significantly improved the growth, survivability, and post-larvae condition of European lobsters compared to the control group. The effect was elevated by combining probiotic (*Bacillus* spp.) and mannan oligosaccharides (MOS) as a diet, where it significantly improved the weight gain, carapace length, length–weight ratio, specific growth rate (SGR), food conversion ratio (FCR), and post-larval condition compared to the group only fed with *Bacillus* spp. or MOS. Based on the current trend, researchers are working on isolating probiotic strains from the GI tract of spiny lobsters to address various diseases and to minimise the mortalities of spiny lobsters during larval stages.

Goulden et al. [127] shortlisted two potential probiotics (*Vibrio* sp. PP05 and *Pseudoalteromonas* sp. PP107) out of 500 marine bacteria, which demonstrated antagonistic effects toward *Vibrio owensii* (pathogen causing epizootics). *Vibrio* sp. PP05 and *Pseudoalteromonas* sp. PP107 also reflected significant protection against the phyllosomas of ornate spiny lobsters when challenged with *V. owensii*, and the survivability was not significantly different compared to the unchallenged group. Nguyen et al. [29] isolated and characterised the bacteria possessing bacteriocin-like activities from lobsters, tiger shrimps, snubnose

pompanos, and cobia as potential probiotics for ornate spiny lobster juveniles against *Vibrio owensii*. Based on in vitro testing, the cell-free supernatants of *Proteus* spp., *Enterococcus faecalis*, *Bacillus cereus*, and *Bacillus pumilus* inhibited the activities of *V. owensii*, thereby indicating the potential for using these bacteria as probiotics in lobster culture. For in vivo testing, *B. pumilus*, *B. cereus*, and *Lactobacillus plantarum* were added to the feed of juvenile spiny lobsters challenged with *V. owensii*. Resultantly, the juvenile spiny lobsters fed with these probiotics demonstrated promising growth, survivability increment, and lower feed conversion rate.

Apart from being significant pathogens of various diseases, *V. owensii* and *V. harveyi* are also commonly detected in cultured shrimps and spiny lobsters [139,140]. In spiny lobster cultures, *V. harveyi* is responsible for the infection of spiny lobsters with luminous vibriosis, causing high mortality rates up to 75% [141]. The introduction of *L. plantarum* as a probiotic could increase the resistance to *V. harveyi* infection by modifying the composition of lactic bacteria and reducing the *V. harveyi* bacteria count in the haemolymph and GI tract [142]. Likewise, the administration of *Bacillus subtilis* as a probiotic in a rearing environment has proven to increase weight gain, SGR, FCR, and enzymatic activities and reduce mortalities [143]. Thus, the use of bacteria strains from the genus *Bacillus* and *Lactobacillus* as probiotics to control disease outbreak in lobster hatchery aquacultures is important and has significantly improved and matured in recent years [144].

## 8. Conclusion and Future Prospect

Given the increasing trend of the present research topic in microbiology, the communities and function of GI microbiota have been studied extensively in humans, mammals, and aquatic vertebrates to ameliorate health and overall conditions. Studies of the GI microbiota in aquatic invertebrates mainly focus on aquaculture species, such as Pacific white shrimps (*Litopenaeus vannamei*), black tiger shrimps (*Penaeus monodon*), and giant freshwater prawns (*Macrobracium rosenbergii*). Meanwhile, the GI microbial function and characteristics are still underreported in a wide range of aquatic invertebrates species, such as spiny lobsters. Moreover, research on the composition of microflora in the GI tract of aquatic invertebrates focuses mainly on bacteria. Some studies revealed that fungi, yeast, and viruses concurrently inhabit the GI tract of spiny lobsters. The advantages of GI bacteria have been proven in several studies, but fungi and yeast inhabiting the GI tract are often overlooked. Recent studies revealed that some marine fungi and yeast inhabit the GI tract of aquatic vertebrates, and invertebrates possess antimicrobial properties. Thus, more studies are needed to elucidate the function and the fungus–host interaction in the GI tract of aquatic vertebrates and invertebrates.

Future research should also consider the interaction between spiny lobster GI microbiota and its host. The GI microbiota has become an essential factor in the success of the spiny lobster culture, as it plays important biological functions, such as nutrition and disease resistance. An extensive research gap has been identified in spiny lobster GI microbial studies from the early 20th to 21st centuries. The focus of GI microbiota studies resumed in the last decades to exploit the probiotic possibilities in spiny lobster aquaculture and to provide sufficient nutrition and enhance their survivability in larval stages. Studies relating to the important underlying mechanisms of indigenous GI microbial communities towards the host, the host–microbes relation in the GI microbiota of spiny lobsters, the effect the GI microbiota has on the host, and more mature and advanced probiotic approaches to the spiny lobster are important in future GI microbe research.

**Author Contributions:** E.Y.L., A.D.T. and M.T.M.L. contributed equally to the first draft of the manuscript. B.A.V.M., K.H., M.S. and C.L.C. contributed to the further revisions of the manuscripts. All authors have read and agreed to the published version of the manuscript.

**Funding:** E.Y.L was funded by the Universiti Malaysia Sabah through the Postgraduate Research Grant (GUG0451-1/2020) and Enrich Marine Sdn. Bhd. through the External Grant Scheme with Universiti Malaysia Sabah (GLS0018-2018). M.T.M.L acknowledges the support from the Tokyo Uni-

versity of Marine Science and Technology, Japan, through the JSPS Core-to-Core Program, "Building up an international research network for successful seed production technology development and dissemination leading South-East Asian region".

**Institutional Review Board Statement:** Not applicable.

**Data Availability Statement:** Not applicable.

**Acknowledgments:** We thank two anonymous reviewers and the academic editors of Fishes for their helpful comments and suggestions that greatly improved an earlier version of this manuscript.

**Conflicts of Interest:** The authors declare no conflict of interest.

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
