# Peer review of "Gastrointestinal Microbiota of Spiny Lobster: A Review"

_fishes, doi:10.3390/fishes7030108_

Round 1

Reviewer 1 Report

I read carefully the reviewed manuscript and observed that the authors made significant revision for their initial one. The current version fits well with the journal and provide valuable information for the MDPI journal readers. I believe that this review will give attention from the readers.

Author Response

We thank and appreciate the reviewer's positive feedback

Reviewer 2 Report

The manuscript has been improved. However a further reading is required for minor corrections:

line 136: Photobacterium

line 166: bacterial cells

line 231: Vibrio composition

line 256: nitrogen fixing bacteria are the .....

line 270: Flavobacterium

line 332: gut associated lymphoid (GALT)

line 386-390; 398-399 : Italics: Firmicues Actinobacteria, Baacteroidetes, Proteobacteria, Tenericutes

line 439: squat lobster fed....

line 474: the enzymes synthesised by GI.......

line 476-478: Lactobacillus

line 479: are clinically proven.....

line 488: European lobster

line 504: in vitro test

line 517: Bacillus subtilis

line 519:  the use of bacterial strains from genus Bacillus and Lactobacillus

line 542: ..as it plays ....

line 544: microbial studies from..... 

Author Response

We thank you for the comprehensive and positive review of our manuscript. The comments were very constructive, and we tried to address all of the concerns.

This manuscript is a resubmission of an earlier submission. The following is a list of the peer review reports and author responses from that submission.

Round 1

Reviewer 1 Report

The submitted manuscript with the manuscript ID: fishes-1553108 entitled 'Gastrointestinal Microbiota of Spiny Lobster: A Review', written by En Yao Lein, Mohammad Tamrin Mohamad Lal, Balu Alagar Venmathi Maran, Choon Looi Ch’ng, Katsuyuki Hamasaki, Motohiko Sano, Audrey Daning Tuzan, submitted to the MDPI Journal Fishes to section: Welfare, Health and Disease, presents a review about gastrointestinal microbiota. It is a scoping literature review which attempts to elucidate gastrointestinal microbiota of spiny lobster with the focus on probiotics, roles of microbiota in digestion, nutrition absorption, and immune systems. It also informs external factors which may drive microbiota. Although, it presents a comprehensive set of references with a coherent flow of story, the manuscript still lacks its critical author’s arguments and stringent formulation of gap of knowledge. Moreover, the manuscript looks like a summary of previous studies rather than informing a new insight of gastrointestinal microbiota of spiny lobster. This can be seen for example by citing the study of other crustaceans, i.e. crab, Portunus trituberculatus (in line 443-452).

The conclusion is too broader, especially if the authors want to present their idea about methods and strategies for better diseases control, health management, effective growth and closing life cycle of spiny lobster.

The manuscript can be improved for example by avoiding repetition information from cited references, but mostly to what the authors can learn from previous findings and then implement/adopt those findings for the case of spiny lobster. These involve in term of colonization mechanism of microbial communities towards the host, interaction of host-microbes relation in in different life-stages of spiny lobster, and potential application of advanced probiotic approaches on spiny lobster. Also, the manuscript needs to emphasize welfare, health and disease issues to match with the section. 

The manuscript needs major revision.

Reviewer 2 Report

The work to review the development of microbiota study in sping lobster. The work is cover the relative works. I believe authors did a better work.

But there some comment need to answer:

1) there are some reprtitive descriptions, as L31-33, it was similar with L33-35. There is only a example, please check the manuscript.

2)L76, what is means "after a decade", what's year begin?

3) Role of GI microbiota in sping lobster is summary as digestion, nutrition, immune system, please change 6 as 5.1, 7 as 5.2, 8 as 5.3, 9 as 6, 10 as 7, 11 as 8.

The work is better. Overall, I found this article enjoyable to read, and it might serve as a guide for other microbiota of sping lobster.

Reviewer 3 Report

The content  of the submitted manuscript are on the gastrointestinal microbiota of spiny lobster even if the second part of the review is  general on  bacteria and is not focused on the title. In paragraph 7, 8, 9 the review tells about the role of microbiota  not on spiny lobster which is the subject of the review (if the literature lacks on this subject you should say). Moreover in this part of the review there are repetitions over the document.

However the review must be corrected for the English language because there are different grammar  mistakes and the meaning of the sentence is not clear. In addition some species names are incorrected.

Various references of Table 1 are old for a review paper.

Some  of the most significant points to be reconsidered  are marked in the attached pdf:
